# Metabolic flexibility via mitochondrial BCAA carrier SLC25A44 is required for optimal fever

Takeshi Yoneshiro[1,2], Naoya Kataoka[3], Jacquelyn M Walejko[4], Kenji Ikeda[1,5], Zachary Brown[1], Momoko Yoneshiro[1], Scott B Crown[4], Tsuyoshi Osawa[6], Juro Sakai[2,7], Robert W McGarrah[4,8], Phillip J White[4,9], Kazuhiro Nakamura[3], Shingo Kajimura[1,10]*

[1]Diabetes Center and Department of Cell and Tissue Biology, University of California, San Francisco, San Francisco, United States; [2]Division of Metabolic Medicine, Research Center for Advanced Science and Technology, The University of Tokyo, Tokyo, Japan; [3]Department of Integrative Physiology, Nagoya University Graduate School of Medicine, Nagoya, Japan; [4]Duke Molecular Physiology Institute, Duke University School of Medicine, Durham, United States; [5]Department of Molecular Endocrinology and Metabolism, Tokyo Medical and Dental University, Tokyo, Japan; [6]Division of Integrative Nutriomics and Oncology, Research Center for Advanced Science and Technology, The University of Tokyo, Tokyo, Japan; [7]Division of Molecular Physiology and Metabolism, Tohoku University Graduate School of Medicine, Sendai, Japan; [8]Department of Medicine, Division of Cardiology, Duke University School of Medicine, Durham, United States; [9]Department of Medicine, Division of EndocrinologyMetabolism and Nutrition, Duke University School of Medicine, Durham, United States; [10]Division of Endocrinology, Diabetes and Metabolism, Beth Israel Deaconess Medical Center, Harvard Medical School, Durham, United States

*For correspondence:
skajimur@bidmc.harvard.edu

Competing interests: The authors declare that no competing interests exist.

**Abstract** Importing necessary metabolites into the mitochondrial matrix is a crucial step of fuel choice during stress adaptation. Branched chain-amino acids (BCAAs) are essential amino acids needed for anabolic processes, but they are also imported into the mitochondria for catabolic reactions. What controls the distinct subcellular BCAA utilization during stress adaptation is insufficiently understood. The present study reports the role of SLC25A44, a recently identified mitochondrial BCAA carrier (MBC), in the regulation of mitochondrial BCAA catabolism and adaptive response to fever in rodents. We found that mitochondrial BCAA oxidation in brown adipose tissue (BAT) is significantly enhanced during fever in response to the pyrogenic mediator prostaglandin $E_2$ ($PGE_2$) and psychological stress in mice and rats. Genetic deletion of MBC in a BAT-specific manner blunts mitochondrial BCAA oxidation and non-shivering thermogenesis following intracerebroventricular $PGE_2$ administration. At a cellular level, MBC is required for mitochondrial BCAA deamination as well as the synthesis of mitochondrial amino acids and TCA intermediates. Together, these results illuminate the role of MBC as a determinant of metabolic flexibility to mitochondrial BCAA catabolism and optimal febrile responses. This study also offers an opportunity to control fever by rewiring the subcellular BCAA fate.

## Introduction

Eukaryotic cells utilize metabolites in distinct organelles during adaptation to stress. Such adaptive mechanisms provide another layer of flexibility in metabolite utilization, *a.k.a.,* metabolic flexibility, thereby allowing for robust adaptive fitness to stress. For example, leucine is a branched-chain amino acid (BCAA) that is essential amino acid needed for anabolic processes, including protein synthesis and nutrition sensing via mTOR activation; however, leucine is also imported into the mitochondrial matrix for catabolic processes in metabolic organs, such as skeletal muscle and brown adipose tissue (BAT) upon exercise and cold adaptation, respectively (*Neinast et al., 2019a*; *Lynch and Adams, 2014*). The factors controlling the subcellular metabolite utilization, for example, the anabolic vs. catabolic responses to BCAA, during stress adaptation remain poorly understood.

One of the key determinants for subcellular metabolite utilization is mitochondrial carrier proteins. Relative to the outer membrane, the mitochondrial inner membrane is highly impermeable to molecules and ions, and thus, a variety of carrier proteins in the mitochondrial inner membrane play pivotal roles in the regulation of metabolite delivery into the mitochondrial matrix. For instance, mitochondrial pyruvate carrier proteins (MPC1 and MPC2) are responsible for importing pyruvate into the matrix (*Bricker et al., 2012*; *Herzig et al., 2012*). Loss of MPC leads to reduced pyruvate oxidation, thereby shifting cellular metabolism toward glycolysis, *a.k.a.,* the Warburg effect, whereas ectopic expression of MPC1/2 promotes mitochondrial pyruvate oxidation via pyruvate dehydrogenase (PDH) (*Schell et al., 2014*; *Vacanti et al., 2014*). Of note, mitochondrial pyruvate uptake via MPC is essential for optimal adaptation to cold environment because genetic loss of MPC1 reduces glucose oxidation and TCA intermediates in brown adipocytes, leading to impaired BAT thermogenesis and cold tolerance in mice (*Panic et al., 2020*).

We recently identified SLC25A44 as a mitochondrial BCAA carrier (MBC) that is required for mitochondrial BCAA import and subsequent oxidation of BCAA in the matrix (*Yoneshiro et al., 2019*). CRISPRi-mediated depletion of *Slc25a44* decreases mitochondrial BCAA oxidation, thereby attenuating systemic BCAA clearance and cold tolerance in mice (*Yoneshiro et al., 2019*). A limitation of the previous study, however, is that MBC is widely expressed in the brain, skeletal muscle, and BAT, and thus, the tissue-specific contribution of MBC remains unknown. Accordingly, the present study generated the BAT-specific MBC knock-out (KO) mice and rigorously determined the extent to which mitochondrial BCAA catabolism via MBC is required for thermogenesis and metabolic adaptation to fever.

Fever is a physiological response to pyrogens, and in general, associated with a survival benefit. However, prolonged thermogenesis that chronically exceeds heat loss, such as psychological stress-induced hyperthermia, malignant hyperthermia, brain injury-induced fever, and endocrine fever, is deleterious and can cause widespread damage at cellular and organismal levels (*Walter et al., 2016*; *Evans et al., 2015*). Although recent studies uncovered the central neural circuits that control fever in response to pyrogens and psychological stress (*Morrison and Nakamura, 2019*), cellular and molecular mechanisms of febrile responses in peripheral organs are insufficiently understood. Hence, better understandings of the fever-associated metabolic adaptation may offer an effective intervention to soothe persistent fever.

Here, we report that mitochondrial BCAA oxidation in BAT is significantly enhanced during fever triggered by prostaglandin $E_2$ ($PGE_2$) and psychological stress. Mitochondrial BCAA uptake via MBC is crucial for cellular metabolic flexibility to catabolize BCAA in the mitochondria: BAT-specific MBC loss blunts mitochondrial BCAA catabolism and $PGE_2$-induced fever in mice. Mechanistically, mitochondrial BCAA uptake via MBC is crucial for the synthesis of mitochondrial amino acids and TCA intermediates. Together, the present study provides new insights into the molecular mechanism by which mitochondrial BCAA transport via MBC regulates metabolic flexibility to BCAA catabolism and adaptive fitness to fever.

## Results

### Fever promotes BCAA oxidation in BAT

We developed an in vivo monitoring system that allowed us to simultaneously record temperature changes in the body core and peripheral tissues of mice in response to thermogenic stimuli (*Figure 1A, B*). As a stimulus of fever, we used intracerebroventricular (ICV) administration of $PGE_2$.

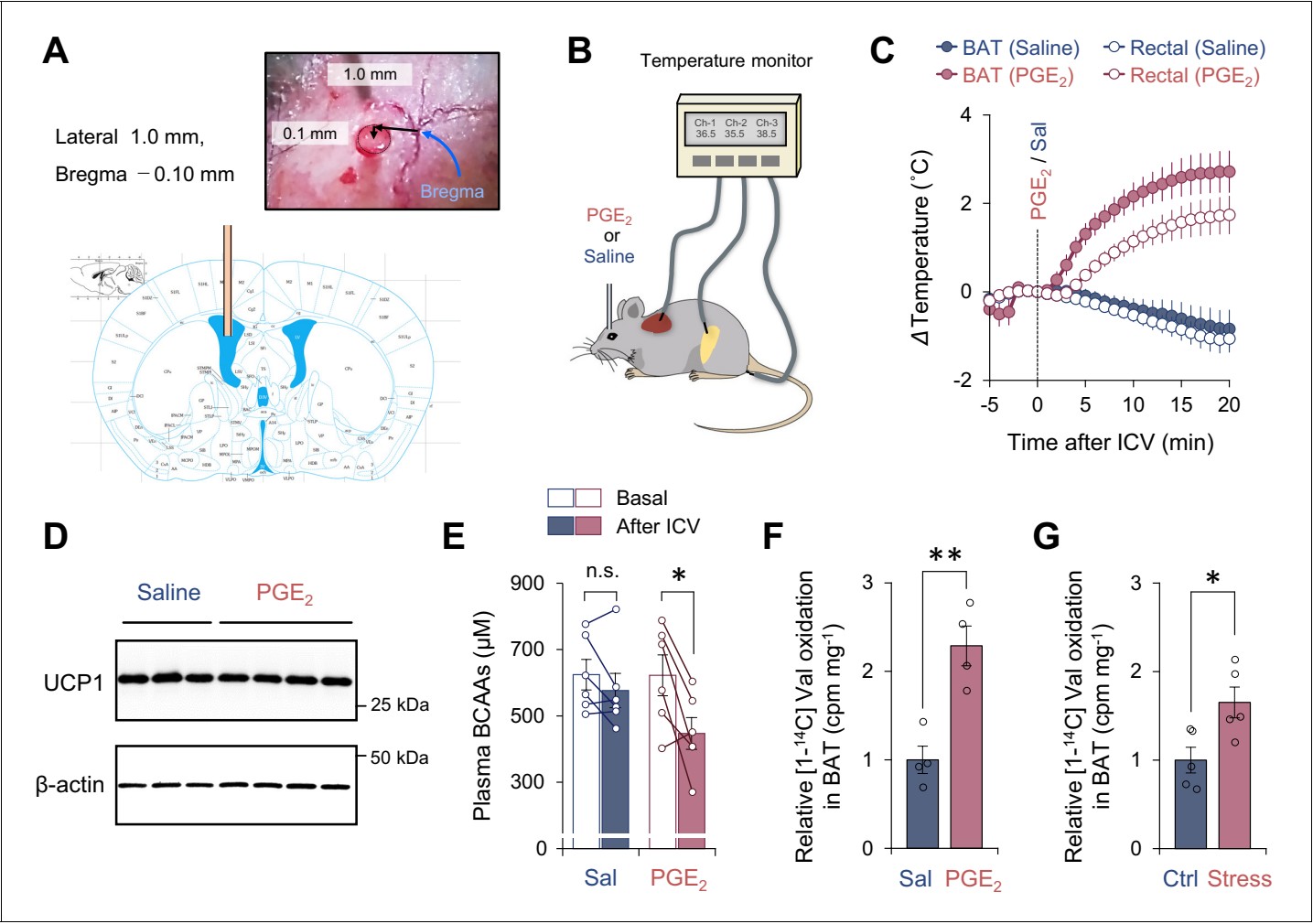

**Figure 1.** Activation of branched chain-amino acid (BCAA) catabolism and thermogenesis in the brown adipose tissue (BAT) following fever stimuli. (**A**) The injection site of prostaglandin E$_2$ (PGE$_2$) in the left lateral ventricle. The image was adapted from 'The mouse brain in stereotaxic coordinates' (*Franklin and Paxinos, 2008*). (**B**) Schematic illustration of the experiment. Mice received intracerebroventricular (ICV) administration of PGE$_2$ (1.4 μg/ mouse) or saline (control). Tissue temperature and rectal temperature were simultaneously recorded by the live monitoring system. (**C**) Real-time temperature changes in the interscapular BAT (iBAT) and rectum of mice in (**B**). $n = 6$ per group. (**D**) Immunoblotting of UCP1 in BAT of mice in (**C**). BAT was harvested after 2 hr of an ICV administration of PGE$_2$. β-Actin was used as a loading control. $n = 3$ for saline, $n = 4$ for PGE$_2$. (**E**) Plasma BCAA concentration before and after an ICV administration of PGE$_2$. Plasma samples were harvested after 2 hr of an ICV administration of PGE2. $n = 6$ per group. (**F**) Relative [1-$^{14}$C] Val oxidation in isolated iBAT of mice. The iBAT tissues of mice were dissected after 2 hr of an ICV administration of PGE$_2$. $n = 4$ per group. (**G**) Relative [1-$^{14}$C] Val oxidation in isolated iBAT of rats exposed to social defeat stress. The iBAT of rats was dissected 2 hr after social defeat stress. $n = 5$ per group. *$p<0.05$, **$p<0.01$, n.s., not significant. Data were analyzed by using paired $t$-test (**E**) or unpaired Student's $t$-test (**F, G**).

The online version of this article includes the following source data for figure 1:

**Source data 1.** Activation of branched chain-amino acid (BCAA) catabolism and thermogenesis in the brown adipose tissue (BAT) following fever stimuli.

PGE$_2$ is a well-established pyrogenic mediator that binds to the EP3 subtype of its receptor in the preoptic area (POA) and triggers the pyrogenic circuit mechanism for the stimulation of sympathetic nerve activity likely by inhibiting thermoregulatory warm-sensitive POA neurons (*Nakamura and Morrison, 2011*; *Nakamura, 2011*). An ICV administration of PGE$_2$ into the POA sufficiently stimulates the non-shivering thermogenesis in BAT without altering the afferent signaling to the POA in rats (*Nakamura et al., 2002*; *Nakamura and Morrison, 2008*). Indicating the importance of the PGE$_2$-mediated signaling in infection-induced fever, EP3 receptor KO mice fail to develop fever in response to intraperitoneal (i.p.) injection of lipopolysaccharide (LPS), an experimental model of systemic bacterial infection (*Ushikubi et al., 1998*; *Lazarus et al., 2007*).

Consistent with previous results (*Nakamura et al., 2002*), an ICV administration of PGE$_2$ (1.4 μg/mouse) rapidly increased the interscapular BAT (iBAT) temperature within 1–2 min following the administration (*Figure 1C*), whereas it did not change the protein expression of UCP1 in the iBAT (*Figure 1D*). This elevation of iBAT temperature was seen selectively to PGE$_2$ but not due to the technical artifact of ICV injection because iBAT temperature was not increased by an ICV administration of saline. The increase in iBAT temperature was followed by a significant increase in core body (rectal) temperature by approximately 2°C (average core body temperature rose to 38.8°C) with a delay of 1–2 min, suggesting that the iBAT is a major heat source for the development of PGE$_2$-induced fever. Of note, chronic treatment with PGE$_2$ analog or ectopic expression of cyclooxygenase (COX) 2, a rate-limiting enzyme in prostaglandin synthesis, promotes beige fat biogenesis in the inguinal WAT (*Vegiopoulos et al., 2010*; *Madsen et al., 2010*). In contrast, the present study employed central administration of PGE$_2$ to acutely provoke febrile responses, and thus, PGE$_2$-induced browning of WAT was negligible in this context unless mice were acclimated to cold prior to PGE$_2$ administration (see *Figure 2*). Following the PGE$_2$-induced febrile response, plasma levels of BCAA were significantly reduced, whereas no significant change was seen after saline administration (*Figure 1E*). Importantly, this reduction in plasma BCAA levels was accompanied by a significant increase in Val oxidation in the isolated BAT (*Figure 1F*). This observation is consistent with our previous finding that cold-induced thermogenesis stimulates BCAA uptake in BAT and systemic BCAA clearance (*Yoneshiro et al., 2019*).

It is worth noting that the increase in BCAA oxidation in BAT is not specific to ICV administration of PGE$_2$ into mice. Psychological stress-induced hyperthermia is another physiological febrile response in which social defeat stress stimulates excitatory neurotransmission from a ventromedial prefrontal cortical site to the dorsomedial hypothalamus, leading to activation of BAT thermogenesis and stress-induced hyperthermia in rats (*Kataoka et al., 2020*). The role of BAT thermogenesis in psychological stress has also been reported in humans (*Robinson et al., 2016*). Following social defeat stress in rats, we observed a significant increase in BCAA oxidation in the isolated BAT of rats under stress (*Figure 1G*). These results suggest that BCAA oxidation and thermogenesis in the BAT are augmented during the development of fever triggered by PGE$_2$ and psychological stress.

## Mitochondrial BCAA catabolism enhances PGE$_2$-induced fever

Cold exposure and the subsequent activation of β$_3$-adrenoceptor (AR) signaling potently stimulates BAT thermogenesis (*Collins, 2011*). Analyses of the public available data of cold-acclimated mice (GSE51080) indicated that chronic adaptation to 15°C for 2 weeks potently stimulates the mitochondrial BCAA catabolic pathway, including the expression of branched chain keto acid dehydrogenase E1 subunit alpha (BCKDHA), dihydrolipoamide branched chain transacylase (DBT), and the mitochondrial BCAA carrier *Slc25a44* (MBC), as well as the expression of BAT thermogenic genes, such as *Ucp1* and *Ppargc1a* (*Figure 2A*). Similarly, chronic administration of a β$_3$-AR agonist (CL-316243, 0.1 mg/kg/day) for 7 days significantly activated the mitochondrial BCAA oxidation pathway in the BAT (*Figure 2B*).

Under these conditions of enhanced mitochondria BCAA oxidation, the PGE$_2$-induced febrile response was significantly potentiated. This enhanced thermogenesis in BAT was accompanied by higher rectal temperature and inguinal WAT temperature following PGE$_2$ injection (*Figure 2C, D*). In BAT, both cold acclimation and CL-316243 administration significantly increased total heat production (Δ°C × min), maximum temperature ($T_{max}$), and the velocity of tissue temperature change (°C min$^{-1}$) in response to an ICV administration of PGE$_2$ (*Figure 2E*).

## Mitochondrial BCAA carrier SLC25A44 is required for PGE$_2$-induced fever

We next determined the extent to which mitochondrial BCAA entry via MBC/*Slc25a44* is required for PGE$_2$-induced febrile response. To this end, we first generated *Slc25a44* floxed mice and then crossed them with *Ucp1*-Cre mice to generate BAT-specific MBC KO mice (*Ucp1*-Cre × *Slc25a44*$^{flox/flox}$) and littermate control mice (*Slc25a44*$^{flox/flox}$) (*Figure 3A, B*, *Figure 3—figure supplement 1A*). On a regular chow diet, BAT-specific MBC deletion did not alter body weight and tissue mass of metabolic organs, including the BAT, liver, WAT, and skeletal muscle (*Figure 3C*, *Figure 3—figure supplement 1B, C*). Morphologically, we found no major difference in brown adipocytes of BAT-

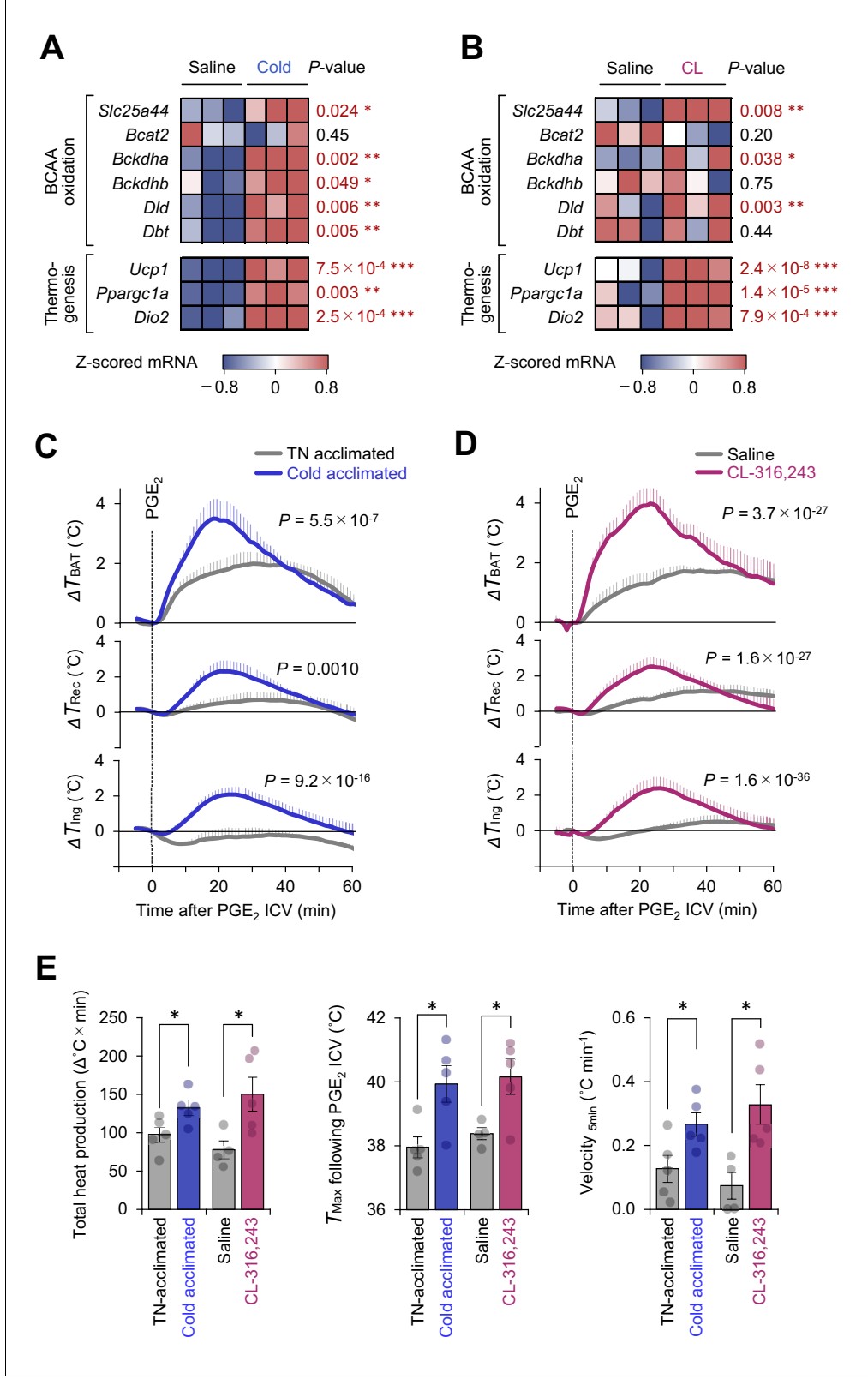

**Figure 2.** Mitochondrial branched chain-amino acid (BCAA) catabolism enhances prostaglandin $E_2$ ($PGE_2$)-induced fever. (**A**) mRNA expression of indicated BCAA catabolism genes and thermogenic genes in brown adipose tissue (BAT) following cold exposure. $n$ = 3 per group. Data were obtained from a previous microarray dataset from GEO under the accession # GSE51080 (***Rosell et al., 2014***). (**B**) mRNA expression of indicated BCAA

*Figure 2 continued on next page*

*Figure 2 continued*

catabolism genes and thermogenic genes in BAT following $\beta_3$-AR agonist (CL-316243; CL) treatment. $n$ = 3 per group. Data were obtained from a previous RNA-sequence dataset from ArrayExpress under the accession # E-MTAB-7445 (*Tajima et al., 2019*). (C) Real-time temperature changes in the interscapular BAT (iBAT) ($\Delta T_{BAT}$), rectum ($\Delta T_{Rec}$), and ing-WAT ($\Delta T_{Ing}$) of mice acclimated to thermoneutrality (TN, 30°C) or cold (15°C) for 2 weeks. $n$ = 5 per group. (D) Real-time temperature changes in the iBAT ($\Delta T_{BAT}$), rectum ($\Delta T_{Rec}$), and ing-WAT ($\Delta T_{Ing}$) of mice treated with $\beta_3$-AR agonist (CL-316243; CL) for 1 week. Saline, $n$ = 4; CL, $n$ = 5. (E) PGE$_2$-stimulated total heat production, maximal temperature ($T_{Max}$), and velocity of tissue temperature in the iBAT of mice in (C) and (D). Data were analyzed by using unpaired Student's $t$-test (A, B, E) or two-way repeated measures ANOVA (C, D). The online version of this article includes the following source data for figure 2:

**Source data 1.** Mitochondrial branched chain-amino acid (BCAA) catabolism enhances prostaglandin E2 (PGE2-)-induced fever.

specific MBC KO mice (*Figure 3D*, *Figure 3—figure supplement 1D*). The phenotype is consistent with our previous study in which neither CRISPRi-mediated knockdown of MBC nor BAT-specific deletion of BCKDHA altered brown adipocyte differentiation (*Yoneshiro et al., 2019*). However, UCP1$^+$ cell-specific loss of MBC significantly reduced valine oxidation by 70% in the BAT (*Figure 3E*), reinforcing the requirement of MBC for mitochondrial BCAA oxidation in vivo. Remaining BCAA oxidation activity in the KO mice may arise from UCP1-negative cells within the BAT, such as endothelial cells, that avoid MBC deletion by *Ucp1*-Cre. No change in BCAA oxidation was seen in the skeletal muscle, a UCP1-negative tissue, of BAT-specific MBC KO mice.

When PGE$_2$ was administered to the BAT-specific MBC KO mice at room temperature, we found that PGE$_2$-induced febrile response was substantially attenuated relative to control mice (*Figure 3F, G*). Specifically, total heat production ($\Delta°C \times min$) and maximum temperature ($T_{max}$) were significantly lower in BAT-specific MBC KO mice than those in control mice (*Figure 3H, I*). Of note, we observed no difference in the velocity of the initial rise in iBAT temperature (°C min$^{-1}$) for the first 5 min after PGE$_2$ administration between the two groups (*Figure 3J*). The data suggest that mitochondrial BCAA uptake via MBC is required for sustaining thermogenic activity in BAT, while it is dispensable for the initiation of thermogenesis following PGE$_2$ administration.

## Mitochondrial BCAA catabolism controls PGE$_2$-induced thermogenesis via UCP1-dependent and UCP1-independent mechanisms

To test if the above phenotype of BAT-specific MBC KO mice is due to reduced BCAA oxidation, we next examined the extent to which genetic loss of BCKDHA, a mitochondria-localized BCAA oxidation enzyme, affects the PGE$_2$-induced febrile response. We previously reported that BAT-specific BCKDHA KO mice (*Ucp1*-Cre × *Bckdha*$^{flox/flox}$) exhibited impaired BCAA oxidation, systemic BCAA clearance, and thermogenesis in BAT (*Yoneshiro et al., 2019*). Consistent with the phenotype seen in BAT-specific MBC KO mice, we found that PGE$_2$-induced BAT thermogenesis was attenuated in the BAT-specific BCKDHA KO mice relative to control mice (*Figure 4A*). It is notable that total heat production ($\Delta°C \times min$) and maximum temperature ($T_{max}$) in the BAT were significantly lower in the BCKDHA KO mice than those in control mice (*Figure 4B, C*), whereas no difference was seen in the velocity of the initial BAT temperature change (°C min$^{-1}$) between the genotypes (*Figure 4D*). These results further support the view that mitochondrial BCAA catabolism through the actions of MBC and BCKDH is required for the maintenance of PGE$_2$-induced BAT thermogenesis, but not required for the initiation of thermogenesis.

Cold-induced thermogenesis in iBAT requires UCP1 (*Enerbäck et al., 1997*). This UCP1 dependency in iBAT is attributed to the fact that brown adipocytes have a low ATP synthesis capacity (*Kramarova et al., 2008*), and thus, ATP-dependent (i.e., UCP1-independent) futile cycle, such as Ca$^{2+}$ cycling or creatine cycling (*Ikeda et al., 2017*; *Kazak et al., 2015*) may not compensate for the lack of UCP1. Nonetheless, several studies reported that UCP1 is dispensable for febrile responses to i.p. administration of LPS or interleukin (IL)-1β (*Okamatsu-Ogura et al., 2007*; *Riley et al., 2016*; *Eskilsson et al., 2020*). Accordingly, we examined if ICV administration of PGE$_2$ induces UCP1-dependent thermogenesis in iBAT. In mice kept under a room temperature condition, we found that PGE$_2$-induced thermogenesis was totally blunted in UCP1 KO mice (*Figure 4E*). The data are consistent with the result in cultured brown adipocytes in which BCAA supplementation

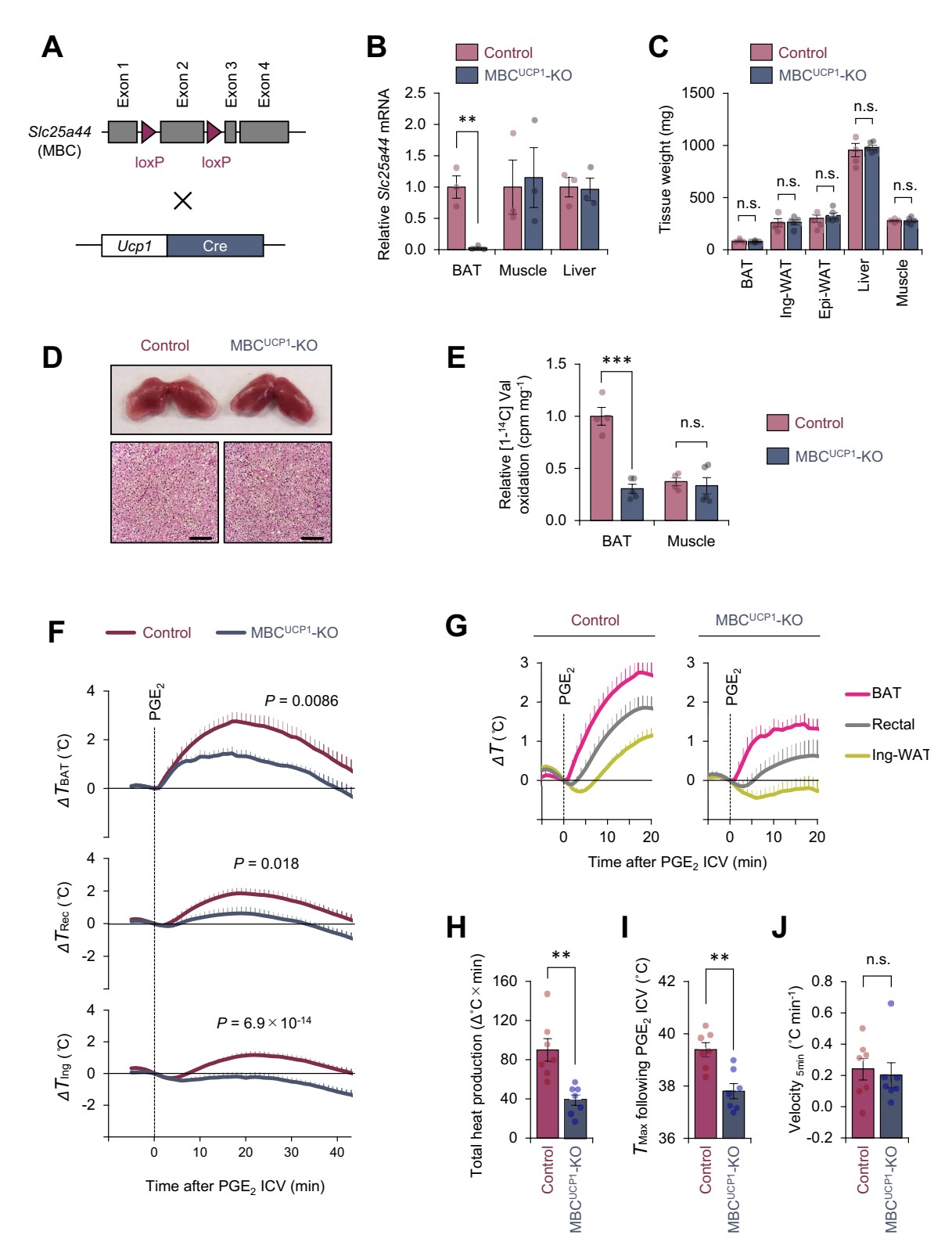

**Figure 3.** Mitochondrial branched chain-amino acid (BCAA) carrier SLC25A44/mitochondrial BCAA carrier (MBC) is required for prostaglandin $E_2$ ($PGE_2$)-induced fever. (**A**) Generation of brown adipose tissue (BAT)-specific MBC knock-out (KO) mice (*Ucp1*-Cre; *Slc25a44*[flox/flox]; MBC[UCP1] KO). (**B**) mRNA expression of *Slc25a44* in the interscapular BAT (iBAT), skeletal muscle, and liver of BAT-specific MBC KO mice and the littermate control mice. $n = 3$ per group. (**C**) Tissue weight of the iBAT, ing-WAT, epi-WAT, liver, and gastrocnemius skeletal muscle of BAT-specific MBC KO mice (MBC[UCP1]

Figure 3 continued

KO) and the littermate control mice. Control, $n$ = 4; MBC$^{UCP1}$ KO, $n$ = 5. (D) Morphology and hematoxylin and eosin (H&E) staining of the iBAT of BAT-specific MBC KO mice and the littermate control mice. Scale bars, 50 μm. (E) Relative [1-$^{14}$C] Val oxidation in the iBAT and gastrocnemius skeletal muscle of BAT-specific MBC KO mice (MBC$^{UCP1}$ KO) and the littermate control mice. Control, $n$ = 4; MBC$^{UCP1}$ KO, $n$ = 5. (F) Real-time temperature changes in the iBAT ($\Delta T_{BAT}$), rectum ($\Delta T_{Rec}$), and ing-WAT ($\Delta T_{Ing}$) of BAT-specific MBC KO mice (MBC$^{UCP1}$ KO) and the littermate control mice. $n$ = 7 per group. (G) Temperature changes during the first 20 min after intracerebroventricular (ICV) administration of PGE$_2$ in (F). (H) Total heat production in the iBAT following ICV administration of PGE$_2$ in (F). (I) $T_{max}$ of iBAT temperature following ICV administration of PGE$_2$ in (F). (J) Velocity of iBAT temperature increase following ICV administration of PGE$_2$ in (F). **p<0.01, ***p<0.001, n.s., not significant. Data were analyzed by using unpaired Student's $t$-test (B, C, E, H–J) or two-way repeated measures ANOVA (F).

The online version of this article includes the following source data and figure supplement(s) for figure 3:

**Source data 1.** Mitochondrial branched chain-amino acid (BCAA) carrier SLC25A44/mitochondrial BCAA carrier (MBC) is required for prostaglandin E2 (PGE2-)induced fever.

**Figure supplement 1.** Characterization of brown adipose tissue (BAT)-specific mitochondrial BCAA carrier (MBC) knock-out (KO) mice.

**Figure supplement 1—source data 1.** Characterization of brown adipose tissue (BAT)-specific mitochondrial BCAA carrier (MBC) knock-out (KO) mice.

enhanced norepinephrine (NE)-stimulated cellular respiration in an UCP1-dependent manner (*Figure 4F*). In contrast, BCAA supplementation in UCP1 KO beige adipocytes significantly enhanced NE-stimulated cellular respiration, although the effect was less than that in wild-type (WT) beige adipocytes (WT: +46.4 ± 1.2%, UCP1 KO: +30.6 ± 2.5%, p<0.001, *Figure 4G*).

Given our previous works on the mechanism of Ca$^{2+}$ cycling thermogenesis via SERCA2 (*Ikeda et al., 2017*), we next determined the extent to which SERCA2-mediated Ca$^{2+}$ cycling mediates BCAA catabolism and thermogenesis. To this end, we depleted SERCA2 (encoded by *Atp2a2*) using two independent shRNAs targeting *Atp2a2* in UCP1 null beige adipocytes (*Figure 4H*). In beige adipocytes, NE treatment significantly stimulated Val oxidation even in the absence of UCP1; however, the NE-induced BCAA oxidation was near completely blunted when SERCA2 was depleted (*Figure 4I*). Importantly, depletion of SERCA2 eliminated the stimulatory effect of BCAA supplementation on UCP1-independent respiration following NE treatment (*Figure 4J*). These results suggest that mitochondrial BCAA oxidation enhances thermogenesis in brown fat and beige fat through UCP1-dependent and independent mechanisms, respectively.

## Metabolic flexibility to mitochondrial BCAA catabolism is required for sustained thermogenesis

Fatty acids and glucose are considered to be the primary fuel for BAT thermogenesis, while the contribution of BCAA as a carbon fuel appears marginal even though BAT is a major metabolic-sink for BCAA (*Yoneshiro et al., 2019*; *López-Soriano et al., 1988*; *Neinast et al., 2019b*; *Hui et al., 2020*). Nonetheless, our data raise the possibility that mitochondrial BCAA catabolism (i.e., mitochondrial import, deamination, and oxidation) is needed for sustaining thermogenic activity even under conditions in which the primary fuels are not limited. To probe this, we cultured differentiated brown adipocytes in the presence of glucose at a physiological level (5 mM) and measured OCR in response to NE. NE treatment rapidly increased OCR within 5–6 min, whereas this effect gradually declined. However, the supplementation of Leu after NE effectively maintained the elevated levels of OCR (*Figure 5A*). On the other hand, such a stimulatory effect of Leu was completely lost in MBC KO brown adipocytes, although these cells were able to respond to NE and increased OCR (*Figure 5B*). The stimulatory effect is not restricted to Leu because supplementation of Val or Ile also significantly augmented norepinephrine-induced thermogenesis to the same extent to Leu in an MBC-dependent manner (*Figure 5C*).

The effect of BCAA supplementation on OCR is to potentiate the action of thermogenic stimuli: when Leu was supplemented in brown adipocytes prior to NE treatment, we observed no increase in OCR. However, NE-induced thermogenesis was significantly enhanced when Leu was supplemented even in the presence of high glucose at 20 mM (*Figure 5D*). Importantly, this effect of Leu was not seen in the absence of MBC (*Figure 5E*). Consistent with the previous reports that activation of the β-AR pathway stimulates both glucose oxidation and lactate production in brown adipocytes (*Olsen et al., 2014*; *Ikeda et al., 2017*), NE treatment in differentiated brown adipocytes increased lactate production. Notably, MBC loss led to higher production of lactate than control cells

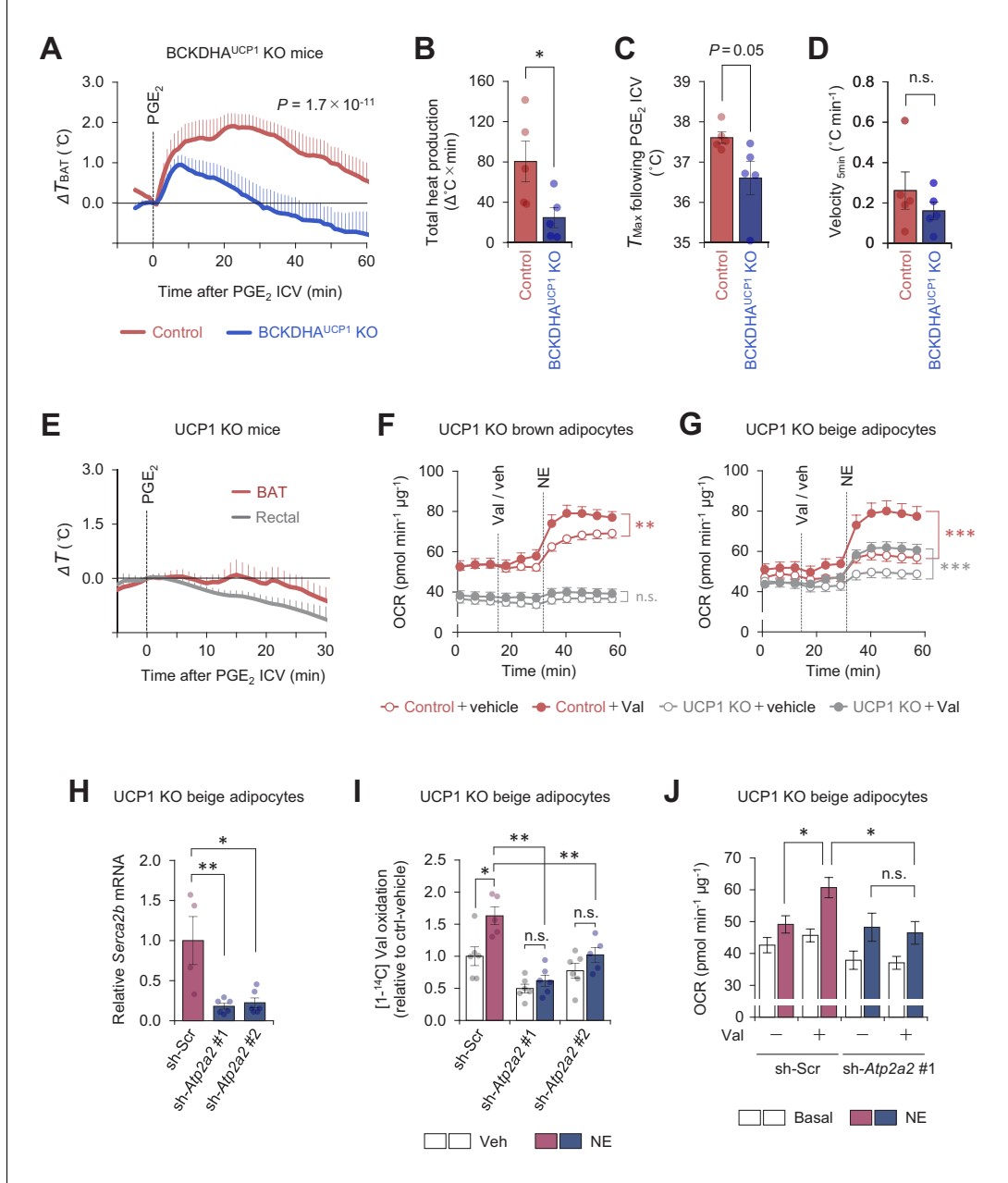

**Figure 4.** Mitochondrial branched chain-amino acid (BCAA) oxidation triggers thermogenesis via UCP1-dependent and UCP1-independent mechanisms. (**A**) Real-time temperature changes in the interscapular BAT (iBAT) of BCKDHA[UCP1]knock-out (KO) and the littermate control mice following intracerebroventricular (ICV) administration of prostaglandin $E_2$ (PGE$_2$). $n = 5$ per group. (**B**) Total heat production in the iBAT of indicated mice following ICV administration of PGE$_2$ in (**A**). (**C**) $T_{max}$ of iBAT temperature following ICV administration of PGE$_2$ in (**A**). (**D**) Velocity of iBAT temperature increase following ICV administration of PGE$_2$ in (**A**). (**E**) Real-time temperature changes in the iBAT and rectum of UCP1 KO mice following ICV injection of PGE$_2$. $n = 5$ per group. (**F**) Oxygen consumption rate (OCR) in brown adipocytes derived from wild-type mice (control) or UCP1 KO mice. Differentiated adipocytes in the BCAA-free medium were treated with valine (Val) or vehicle (Veh), and subsequently stimulated with norepinephrine (NE) at indicated time points. OCR values were normalized by total protein (μg). $n = 10$ per group. (**G**) OCR in beige adipocytes derived from wild-type mice (control) or UCP1 KO mice. Differentiated adipocytes in the BCAA-free medium were treated with valine (Val) or vehicle (Veh), and subsequently stimulated with NE at indicated time points. OCR values were normalized by total protein (μg). Control + vehicle, $n = 9$; Control + Val $n=8$; UCP1 KO + vehicle, $n = 6$; UCP1 KO + Val, $n = 9$. (**H**) mRNA expression of *Serca2b* in UCP1 KO beige adipocytes expressing a scrambled control (Scr, $n = 4$) or two independent shRNAs targeting *Atp2a2* (#1 and #2, $n = 6$ per group). (**I**) Relative [1-14C] Val oxidation in UCP1 KO beige adipocytes in (**H**). Differentiated adipocytes were cultured in the presence of NE or vehicle. Valine oxidation was measured using [1-14C] valine and normalized by total protein (μg). Vehicle $n = 6$, NE $n = 5$, except sh-*Atp2a2* #1 treated with NE $n = 6$. (**J**) OCR was measured in UCP1 KO beige adipocytes in (**H**). Differentiated adipocytes were incubated in the BCAA-free media supplemented with Val (sh-Scr, $n = 9$; sh-*Atp2a2*, $n = 6$) or vehicle (sh-Scr, $n = 6$; sh-

*Figure 4 continued*

*Atp2a2*, *n* = 7), and subsequently stimulated with NE. OCR values were normalized by total protein (μg). *p<0.05, **p<0.01, ***p<0.001, n.s., not significant. Data were analyzed by unpaired Student's *t*-test (B–D), one- (H) or two-way (I) factorial ANOVA followed by Tukey's post hoc test, or two-way repeated measures ANOVA (A, F, G) followed by Tukey's post hoc test (J).

The online version of this article includes the following source data for figure 4:

**Source data 1.** Mitochondrial branched chain-amino acid (BCAA) oxidation triggers thermogenesis via UCP1-dependent and UCP1-independent mechanisms.

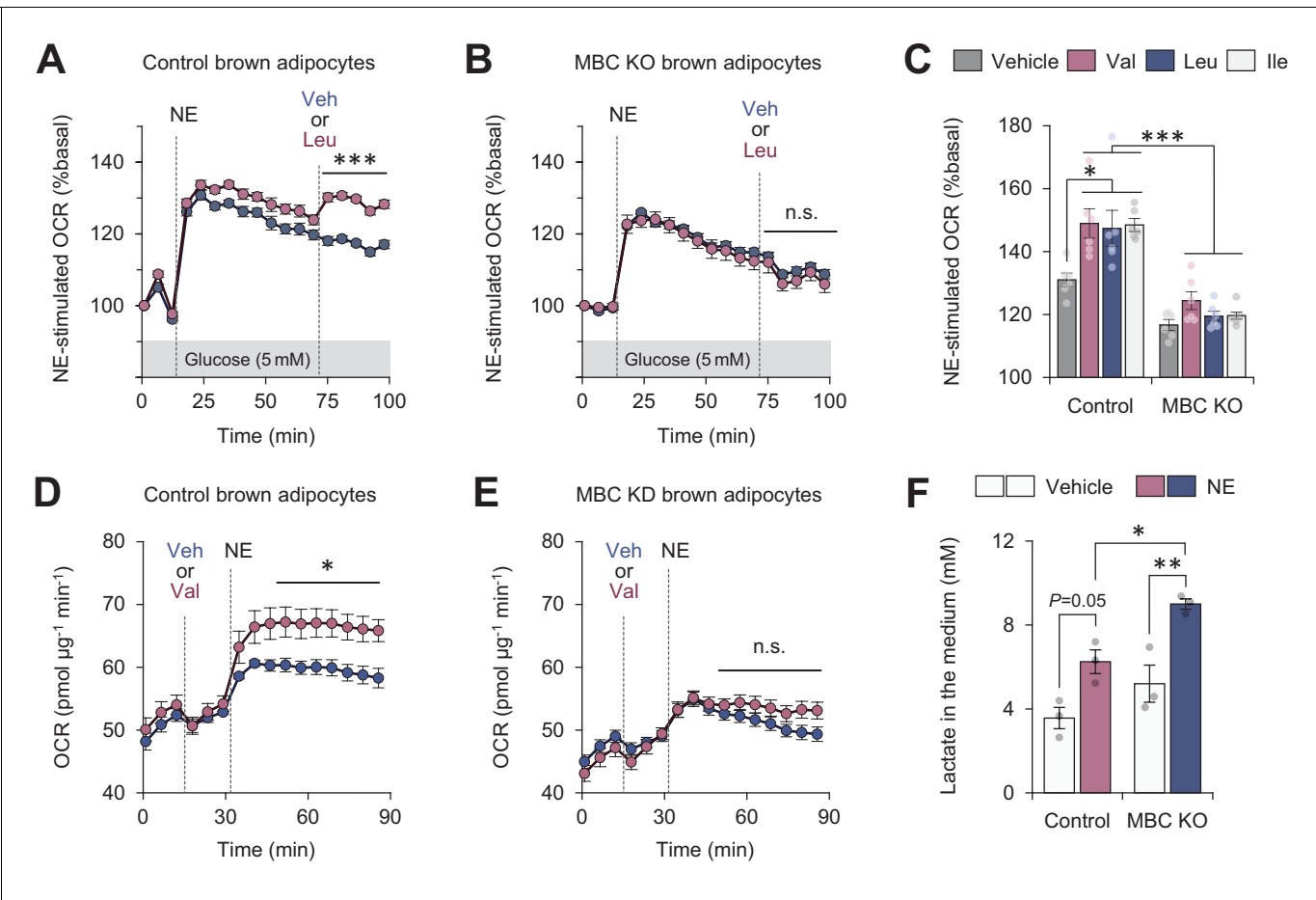

**Figure 5.** Mitochondrial branched chain-amino acid (BCAA) catabolism required for sustained thermogenesis. (A) Oxygen consumption rate (OCR) normalized to total protein (in μg) in control brown adipocytes. Differentiated adipocytes in BCAA-free medium with 5 mM glucose were stimulated with norepinephrine (NE), and subsequently supplemented with Leu or vehicle. *n* = 9 per group. (B) OCR normalized to total protein (in μg) in mitochondrial BCAA carrier (MBC) knock-out (KO) brown adipocytes. Differentiated adipocytes cultured in BCAA-free medium with glucose (5 mM) were stimulated with NE, and subsequently supplemented with Leu or vehicle. *n* = 9 per group. (C) NE-stimulated OCR in control and MBC KO brown adipocytes. Differentiated adipocytes in BCAA-free medium, containing 5 mM glucose, supplemented with Val, Leu, or Ile were stimulated with NE for 30 min. *n* = 6 per group. (D) OCR normalized to total protein (in μg) in control brown adipocytes. Differentiated adipocytes in BCAA-free medium with glucose (20 mM) were supplemented with Leu or vehicle, and subsequently stimulated with NE. *n* = 5 per group. (E) OCR normalized to total protein (in μg) in MBC KO brown adipocytes. Differentiated adipocytes in BCAA-free medium with glucose (20 mM) were supplemented with Leu or vehicle, and subsequently stimulated with NE. *n* = 5 per group. (F) Lactate production in control and MBC KO brown adipocytes. Differentiated adipocytes were stimulated with vehicle or NE for 30 min, and lactate content in the medium was measured. *n* = 3 per group. *p<0.05, **p<0.01, ***p<0.001, n.s., not significant. Data were analyzed by two-way repeated measures ANOVA (A, B, D, E) or two-way factorial ANOVA followed by Tukey's post hoc test (C, F).

The online version of this article includes the following source data for figure 5:

**Source data 1.** Mitochondrial branched chain-amino acid (BCAA) catabolism required for sustained thermogenesis.

(*Figure 5F*), suggesting that MBC KO cells were limited to utilize glucose even in the presence of Leu.

## MBC is required for the synthesis of the mitochondrial amino acid pool and TCA intermediates

To address how MBC controls BCAA catabolism, we next performed uniformly $^{13}$C-labeled, $^{15}$N-labeled Leu ([U-$^{13}$C$_6$, $^{15}$N$_1$] Leu) tracing in differentiated brown adipocytes (*Figure 6—figure supplement 1A*). As we previously reported (*Yoneshiro et al., 2019*), brown adipocytes dominantly express BCAA aminotransferase 2 (BCAT2), the mitochondria-localized form of BCAT, and thus, deamination and oxidation of BCAA largely occur in the mitochondrial matrix (*Figure 6A*). GC-MS analysis of intracellular metabolites revealed that levels of Leu, α-ketoisocaproate (KIC), Ala, and Glu, were significantly lower in MBC KO brown adipocytes than those in WT control cells (*Figure 6B*). There was a trend towards lower Asp levels in MBC KO cells relative to WT control cells, although the difference was not statistically significant. This is likely because BAT expresses very low levels of aspartate aminotransferase (GOT), the enzyme that catalyzes the conversion of oxaloacetate to Asp, whereas BAT expresses high levels of glutamate-pyruvate transaminase (GPT) that catalyzes the reversible conversion of Ala and α-ketoglutarate (α-KG) to Glu and pyruvate (*Figure 6—figure supplement 1B*). Importantly, the reduced levels of KIC, Glu, and Ala in MBC KO cells were due to the reduced conversion from Leu because levels of labeled KIC, Glu, and Ala (i.e., Leu-derived) were significantly lower in KO cells than those in WT control cells (*Figure 6C, D*). Also, there was a slight but significant reduction in M3-labeled Asp in KO cells relative to control cells.

Following the deamination of Leu in the mitochondria, KIC fuels the TCA cycle in brown adipocytes. Our previous study showed that NE treatment significantly increased the Leu-derived TCA cycle intermediates (*Yoneshiro et al., 2019*). Consistent with this prior observation, we found that intracellular levels of α-KG, succinate, malate, and citrate increased after NE treatment; however, the increases in α-KG and malate levels occurred at significantly less degree in MBC KO cells (*Figure 6E*). Importantly, the total pool size of TCA cycle intermediates in MBC KO cells was smaller than that in WT control cells at 60 min after NE treatment (*Figure 6F*), although the differences were not seen at basal state (before NE treatment) (*Figure 6—figure supplement 1C*). These results are likely relevant to the finding that mitochondrial BCAA catabolism is required for sustaining BAT thermogenesis, but not for the initial trigger of thermogenesis. Even under the reduced pool size, the contribution of Leu into the TCA cycle, as shown by $^{13}$C ion counts and the mole percentage enrichment (MPE), was lower in MBC KO cells than that in WT control cells (*Figure 6G, H*). Of note, the MPE of TCA intermediates was lower than our previous study that achieved ~20% (*Yoneshiro et al., 2019*). This difference arose from the fact that the present study incubated cells with $^{13}$C-labeled Leu for 30 min and subsequently washed off before NE treatment, whereas the previous study used a protocol in which cells were labeled for 1 hr during NE treatment.

Together, the present data suggest the following model (*Figure 7*). In response to febrile stimuli, such as increased PGE$_2$ or psychological stress, mitochondrial BCAA catabolism is activated along with enhanced BAT thermogenesis. The sympathetically stimulated mitochondrial BCAA catabolism is required for maintaining active BAT thermogenesis to sustain febrile responses. Mechanistically, MBC is a critical gatekeeper of mitochondrial BCAA transport and subsequent catabolic reactions, including deamination as well as the synthesis of mitochondrial amino acid pool and TCA intermediates.

## Discussion

In general, hyperthermic responses to pyrogens are considered beneficial because the elevation of body temperature can prevent the replication of infective pathogens, while activating innate immunity (e.g., enhanced cytokine release). However, persistent thermogenesis, often associated with psychological stress-induced hyperthermia, malignant hyperthermia, traumatic brain injury, and endocrine fever, causes severe exhaustion and, in the worst cases, leads to widespread cellular and organ damage, such as renal injury, liver failure, and edema (*Walter et al., 2016*).

Fever involves multiple involuntary effector responses besides BAT thermogenesis, including skeletal muscle shivering and cutaneous vasoconstriction (*Morrison and Nakamura, 2019*). Hence, experimental setting, such as the types of pyrogen and anesthetizer, would influence the data

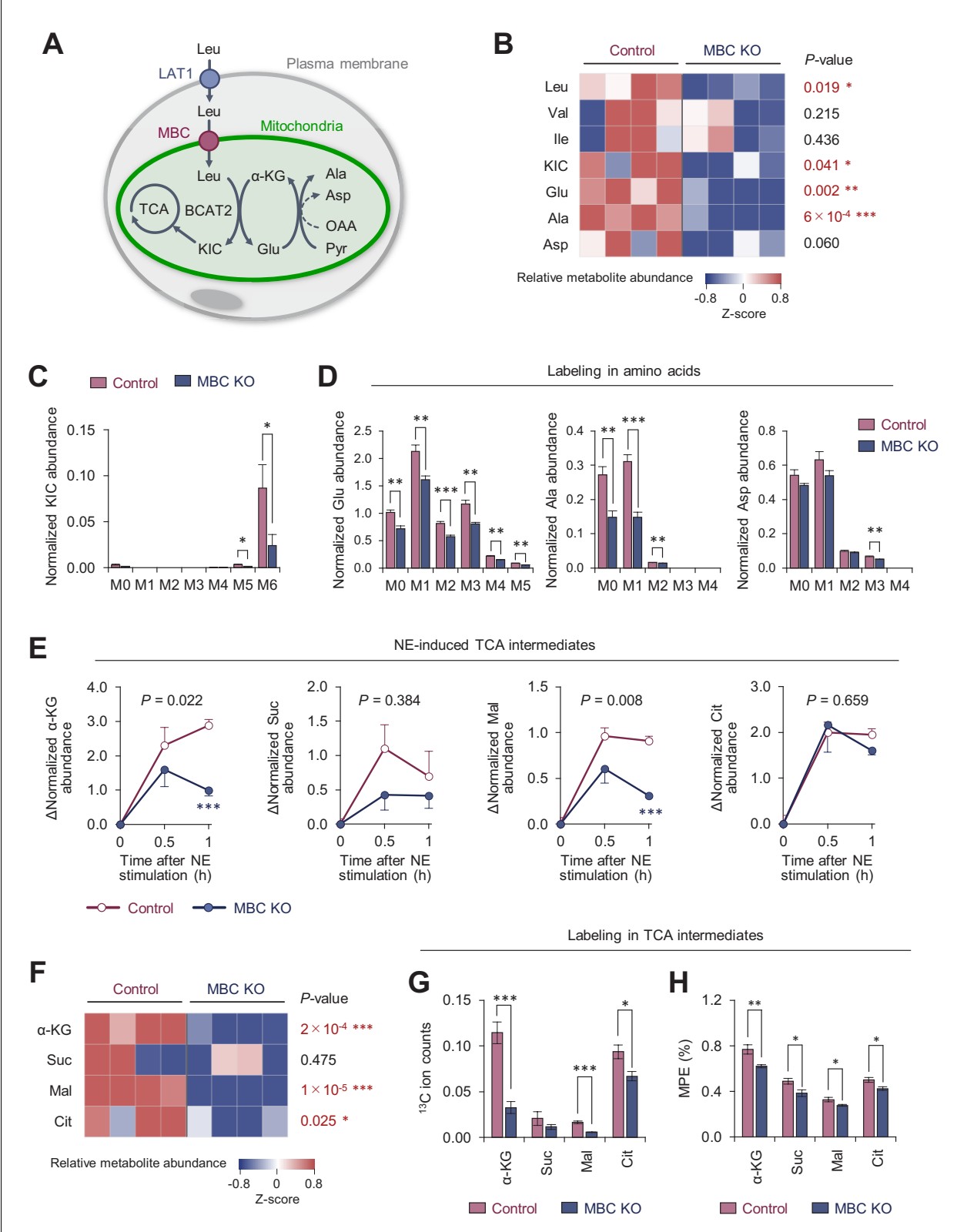

**Figure 6.** Mitochondrial BCAA carrier (MBC) is required for the synthesis of the mitochondrial amino acids and TCA intermediates. (A) Schematic illustration of mitochondrial branched chain-amino acid (BCAA) fate to the TCA cycle and amino acid synthesis. (B) Amino acid relative abundance in wild-type (WT) control and MBC knock-out (KO) brown adipocytes as determined by [13]C-Leu metabolomics. Cells were harvested at a basal state. *n* = 4 per group. (C) Leu-derived [13]C labeling in KIC in WT control and MBC KO brown adipocytes in (B). (D) Labeling in indicated amino acids derived from

*Figure 6 continued on next page*

*Figure 6 continued*

Leu in WT control and MBC KO brown adipocytes in (**B**). (**E**) Norepinephrine (NE)-induced changes in TCA cycle intermediates in WT control and MBC KO brown adipocytes. Cells were harvested at the indicated time points following NE treatment in (**B**). (**F**) The relative abundance of indicated TCA cycle intermediate in WT control and MBC KO brown adipocytes at 1 hr following NE treatment in (**B**). (**G**) Leu-derived $^{13}$C labeling in TCA cycle intermediates in WT control and MBC KO brown adipocytes. Cells were harvested at 1 hr following NE treatment in (**B**). (**H**) Mole percentage enrichment (MPE) of TCA cycle intermediates derived from [U-$^{13}$C$_6$] Leu in control and MBC KO brown adipocytes. Cells were harvested at 1 hr following NE treatment. $n$ = 4 per group. Data in 6B-H was normalized to a norvaline internal standard. *p<0.05, **p<0.01, ***p<0.001. Data were analyzed by unpaired Student's *t*-test (**B–D, F–H**) or two-way factorial ANOVA followed by post hoc unpaired *t*-test (**E**).

The online version of this article includes the following source data and figure supplement(s) for figure 6:

**Source data 1.** Mitochondrial BCAA carrier (MBC) is required for the synthesis of the mitochondrial amino acids and TCA intermediates.
**Figure supplement 1.** Metabolic fate of Leu in mitochondrial BCAA carrier (MBC) knock-out (KO) brown adipocytes.
**Figure supplement 1—source data 1.** Metabolic fate of Leu in mitochondrial BCAA carrier (MBC) knock-out (KO) brown adipocytes.

interpretation. For instance, UCP1 null mice exhibit fever in response to systemic administration (i.p. injection) of LPS or IL-1β under a non-anesthetized condition (*Okamatsu-Ogura et al., 2007*; *Riley et al., 2016*; *Eskilsson et al., 2020*). The difference from the present study is due to the fact that, at room temperature, BAT thermogenesis can be compensated by involuntary (e.g., shivering, vasoconstriction) and behavioral (e.g., increased locomotion) responses until body temperature reaches to a set-point in the pyrogen-sensitized thermoregulatory circuitry of the brain (*Naka-mura, 2011*; *Morrison and Nakamura, 2019*). On the other hand, the present temperature record-ing in anesthetized mice excludes such responses and specifically determines the contribution of non-shivering thermogenesis to febrile responses in response to the ICV administration of PGE$_2$. In this regard, the contribution of skeletal muscle to circulating BCAA (e.g., protein breakdown) would be minimum in our study. It is likely that the primary source of circulating BCAAs is food-derived, although microbiome could be an alternative source (*Pedersen et al., 2016*).

Nonetheless, the present study demonstrates that mitochondrial BCAA catabolism plays a vital role in maintaining BAT thermogenesis to sustain febrile responses, and thus suggests a new oppor-tunity to soothe persistent fever by blocking this pathway. Conversely, it is conceivable that enhanc-ing BCAA catabolism is effective for elderly populations in which approximately 20–30% of the elderly show blunted or absent febrile response to pyrogens (*Norman, 2000*). It is worth noting that the thermogenic capacity of BAT is significantly attenuated in aging, and that this attenuation is closely coupled with reduced BCAA oxidation (*Tajima et al., 2019*). This age-associated attenuation in BCAA oxidation is attributed, in part, to reduced protein lipoylation in the E2 subunit of the BCKDH complex, while dietary supplementation of α-lipoic acids effectively restores the protein lip-oylation of BCKDH and thermogenic function of BAT in old mice (*Tajima et al., 2019*). Since a blunted febrile response to infections represents a poorer prognosis in the elderly (*Norman, 2000*), enhancing BCAA catabolism, for example, by α-lipoic acid supplementation or inhibition of BCKDH kinase activity (BDK, an inhibitory kinase of BCKDH), may represent effective means to alleviate age-associated decline in fever responses.

## BCAAs are more than fuel in thermogenic fat

It has been well appreciated that lipolysis-derived fatty acids are the primary fuel for thermogenesis in brown and beige fat (*Chouchani and Kajimura, 2019*). Indeed, blockade of lipolysis in adipose tis-sues, by genetically deleting adipose triglyceride lipase (ATGL) or the ATGL-activating protein com-parative gene identification-58 (CGI-58) in adipocytes, potently attenuates BAT thermogenesis in mice (*Schreiber et al., 2017*; *Shin et al., 2017*). On the other hand, the contribution of BCAA to the TCA cycle appears minor relative to fatty acids or glucose in BAT (*Hui et al., 2020*), although BAT is a significant metabolic sink for BCAA when activated (*Yoneshiro et al., 2019*; *López-Soriano et al., 1988*; *Neinast et al., 2019b*). This is aligned with the present results that mitochondrial BCAA uptake and oxidation are required for the maintenance, rather than the initiation, of BAT thermogen-esis (see *Figures 3* and *4*). Presumably, BAT is able to initiate the rising phase of febrile thermogen-esis by utilizing fatty acids and glucose even in the absence of BCAA.

We speculate the following possibilities as to why mitochondrial BCAA catabolism is required in BAT: The first possibility is the role of BCAA catabolic products as replenishment of TCA cycle inter-mediates. Complete oxidation of acetyl-CoA from β-oxidation requires the condensation of acetyl-

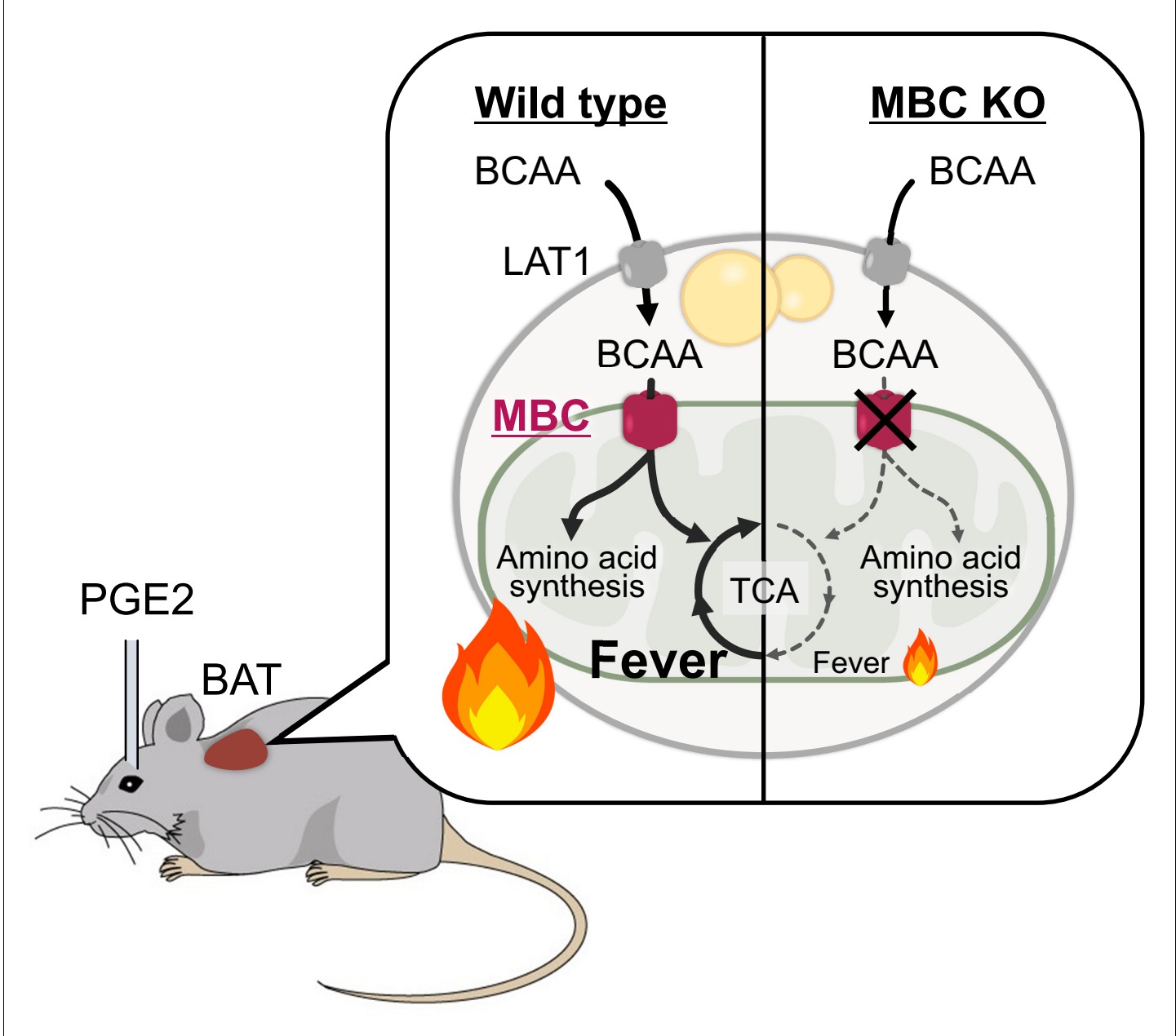

**Figure 7.** A model of mitochondrial branched chain-amino acid (BCAA) catabolism and brown adipose tissue (BAT) thermogenesis in response to fever stimuli. In response to intracerebroventricular (ICV) administration of prostaglandin E$_2$ (PGE$_2$) or psychological stress, mitochondrial BCAA catabolism in the BAT is activated along with enhanced thermogenesis. The fever-induced mitochondrial BCAA catabolism is required for sustaining thermogenesis. Mitochondrial BCAA uptake via mitochondrial BCAA carrier (MBC) is required for BCAA deamination and the synthesis of mitochondrial amino acid pool and TCA intermediates.

CoA and oxaloacetate in the TCA cycle. Generation of the anaplerotic products from BCAA oxidation (see *Figure 6*) may allow for sustaining fatty acid oxidation in the mitochondrial matrix. The second possibility is the role of BCAA-derived nitrogen in the mitochondria. Given the role of BCAAs as major nitrogen donors, deamination of BCAAs by BCAT2 is critical for the synthesis of mitochondrial amino acid pool, including Glu, Ala, and Asp. Indeed, all three BCAAs (Val, Leu, Ile) similarly augmented NE-stimulated respiration in an MBC-dependent manner. Also, thermogenic stimuli activate the purine and pyrimidine synthesis pathway in BAT (*Lu et al., 2017*), and hence, BCAA-derived nitrogen could contribute to nucleotide synthesis. Pertinent to the finding, our recent study reports

that MBC overexpression upregulates the expression of SLC7A5 (LAT1), the plasma membrane transporter for neutral amino acids, including BCAAs (*Walejko et al., 2021*). Third, BCAA catabolism may promote de novo lipogenesis by generating monomethyl branched-chain fatty acids (mmBCFAs) as previously reported (*Green et al., 2016*; *Wallace et al., 2018*). BAT is a major site of mmBCFA synthesis in which the expression of mmBCFA synthesis enzymes, such as carnitine acetyl-transferase (CrAT), is increased after 1-month-cold acclimatization (*Wallace et al., 2018*). This pathway is activated during chronic cold adaptation (3 weeks and after) (*Yoneshiro et al., 2019*), and thus, BCAA may contribute to mmBCFA synthesis under a cold-acclimated state. Together, the role of BCAA in thermogenic fat is likely more than merely a carbon source of fuel into the TCA cycle.

## The role of mitochondrial carriers in metabolic flexibility

Metabolic flexibility is a term that often describes the ability of cells, organs, or an organism to shift the principal types of fuel in order to meet their metabolic demands in response to changes in external and internal conditions. A well-appreciated example is in the skeletal muscle upon exercise in which acute and high-intensity exercise utilizes glucose oxidation, while aerobic and endurance exercise triggers a fuel switch toward fatty acid oxidation (*Smith et al., 2018*; *Goodpaster and Sparks, 2017*). Importing necessary metabolites into the mitochondrial matrix (e.g., acyl-CoA for fatty acids) is an essential step of metabolic flexibility: for instance, blockade of carnitine shuttle by deleting carnitine palmitoyltransferase 1 (CPT1) attenuates muscle function and endurance exercise capacity in mice (*Wicks et al., 2015*). It is pertinent to note that genetic mutations in carnitine-acylcarnitine translocase (encoded by *SLC25A20*), which mediates the transport of acylcarnitines into the mitochondrial matrix, cause severe metabolic disorders, including hypoglycemia, myopathy, and muscle weakness in humans (*Fukushima et al., 2013*; *Iacobazzi et al., 2004*).

The present study found that when stimulated with NE, brown adipocytes lacking MBC increase lactate production, suggesting a shift to glucose utilization even in the presence of BCAA. This metabolic inflexibility is accompanied by an impairment in sustained thermogenic activity in brown adipocytes. Of particular note, impaired BCAA oxidation in peripheral metabolic organs and concomitant elevation of circulating BCAA levels are a metabolic signature of obesity, insulin resistance, and type 2 diabetes (*Felig et al., 1969*; *Wang et al., 2011*; *Newgard et al., 2009*; *Würtz et al., 2013*). Quantitative BCAA tracing shows that reduced BCAA oxidation in the liver and adipose tissues re-distributes BCAA into the muscle, leading to BCAA overload and insulin resistance (*Neinast et al., 2019b*). Thus, future studies are needed to determine the role of MBC in the pathogenesis of insulin resistance.

# Materials and methods

## Animals

All the animal experiments were performed following the guidelines by the UCSF Institutional Animal Care and Use Committee (AN165833) or by the Nagoya University Animal Experiment Committee, and approved by the committees. The WT and UCP1 KO mice aged ~14 weeks had free access to food and water, 12 hr light cycles, and were caged at 23°C. BAT-specific *Bckdha* KO mice were obtained by crossing *Bckdha* floxed mice with *Ucp1*-Cre mice, as reported previously (*Yoneshiro et al., 2019*). For the generation of BAT-specific *Slc25a44*/MBC KO mice (MBC[UCP1] KO mice), *Slc25a44* floxed mice were obtained from the Applied StemCell (MC185, Milpitas, CA, USA) and crossed with *Ucp1*-Cre mice. All mice were C57BL/6 background. Adult male Sprague–Dawley (SD) rats were housed two or three to a cage with ad libitum access to food and water at 24°C with a standard 12 hr light/dark cycle.

## Cold acclimatization and treatment with $\beta_3$ adrenergic receptor agonist

WT mice were acclimated to either thermoneutral temperature at 30°C ($n = 5$) or cold temperature at 15°C ($n = 5$) for 2 weeks and were used for temperature recording in fever response. UCP1 KO mice and the littermate WT mice (control) were treated with saline (control $n = 4$, UCP1 KO $n = 5$) or 0.1 mg/kg/day of CL-316243, $\beta_3$ adrenergic receptor agonist (control $n = 5$, UCP1 KO $n = 6$) for 1 week and were used for temperature recording in fever response.

### Temperature recordings in febrile response

Mice under anesthesia with urethane (1.3 mg/kg) were positioned in a stereotaxic apparatus according to the mouse brain atlas (*Franklin and Paxinos, 2008*). A stainless steel cannula (outer diameter, 0.35 mm) and a 1 ml disposable syringe were connected with polyethylene tubing and were inserted perpendicularly into the right lateral ventricle (coordinates: 0.1 mm posterior to bregma, 0.1 mm lateral to the midline, and 1 mm ventral to the brain surface). Then, mice were left on a self-regulating heating pad to stabilize the rectal temperature ($T_{Rec}$) at 37°C and implanted with a type T thermocouple probe (IT-18, Physitemp, Clifton, NJ) in the rectum and with needle microprobes (MT-29/1, Physitemp) in the interscapular BAT and inguinal WAT (*Yoneshiro et al., 2019*). Tissue temperatures were simultaneously recorded by TC-2000 Meter (Sable Systems International). When tissue temperatures were stable, for >5 min, $PGE_2$ (1.4 µg; Sigma) in 1.4 µl of pyrogen-free saline or only saline was injected into the ventricle through the cannula. The changes in tissue temperatures were contentiously measured every second for 1 hr or as indicated, and the mean value per minute was calculated.

## Tissue histology

For hematoxylin and eosin (H&E) staining, tissues of mice were fixed in 4% paraformaldehyde overnight at 4°C, followed by dehydration in 70% ethanol. After the dehydration procedure, tissues were embedded in paraffin, sectioned at a thickness of 5 µm, and stained with H&E following the standard protocol. Images of tissue samples were captured using the Inverted Microscope Leica DMi8.

### Ex vivo BCAA oxidation

Isolated tissue (20–30 mg) was placed in a polypropylene round-bottom tube and incubated in the 1 ml KRB/HEPES buffer containing 0.16 µCi/ml [1-$^{14}$C] Val at 37°C for 1 hr. After adding 350 µl of 30% hydrogen peroxide in the tube, [$^{14}$C] $CO_2$ was trapped in the center well supplemented with 300 µl of 1 M benzethonium hydroxide solution for 20 min at room temperature. BCAA oxidation was quantified by counting radioactivity of trapped [$^{14}$C] $CO_2$ using a scintillation counter.

### Cell culture

Immortalized brown and beige adipocytes from C57BL/6 mice and UCP1 KO brown and beige adipocytes were generated in our previous studies (*Yoneshiro et al., 2019*; *Ikeda et al., 2017*). Adipocyte differentiation was induced by treating confluent preadipocytes with DMEM containing 10% FBS, 0.5 mM isobutylmethylxanthine, 125 nM indomethacin, 2 µg/ml dexamethasone, 850 nM insulin, 1 nM T3, and 0.5 µM rosiglitazone. Two days after induction, cells were switched to a maintenance medium containing 10% FBS, 850 nM insulin, 1 nM T3, and 0.5 µM rosiglitazone (*Ohno et al., 2012*). The cells were fully differentiated 6–7 days after inducing differentiation.

### Generation of *Slc25a44*/MBC KO brown adipocytes

For the generation of *Slc25a44*/MBC KO brown adipocytes, preadipocytes isolated from BAT of *Slc25a44*$^{flox/flox}$ mice were immortalized by using the SV40 Large T antigen as described previously (*Yoneshiro et al., 2019*) and subsequently infected with a retrovirus containing either empty vector or Cre (#34565, Addgene), followed by hygromycin selection at a dose of 200 µg/ml.

### Cellular respiration

Oxygen consumption rate (OCR) in cultured adipocytes was measured using the Seahorse XFe Extracellular Flux Analyzer (Agilent) in a 24-well plate. For measurement of NE-induced respiration in the presence and absence of BCAA, differentiated adipocytes were maintained either in KRB/HEPES buffer containing 20 or 5 mM glucose, 200 nM adenosine, 2% BSA (bovine serum albumin), or in BCAA-free DMEM (Dulbecco's modified Eagle medium), which was a kind gift from Nissui Pharmaceutical Co. Ltd. During OCR measurement, cells were treated with 2 mM Val, Leu, Ile, or vehicle, and subsequently treated with NE (1 µM) at the indicated time point.

### Stable isotope-labeled leucine metabolome analysis

To determine the effect of MBC depletion on the metabolic fate of BCAA in brown adipocytes, we used [$^{13}C_6$, $^{15}N_1$] Leu tracing. Differentiated MBC KO or control brown adipocytes were incubated in

the BCAA-free medium supplemented with [$^{13}C_6$, $^{15}N_1$] Leu (608068, Sigma-Aldrich) and were harvested at 0, 0.5, and 1 hr after the treatment with NE. Metabolites were then extracted using by scraping cells in 3 ml chilled 100% MeOH (LC-MC grade). The extracts were then centrifuged at 4°C and 14,400 × g for 10 min, and the clarified aqueous phase was transferred to a fresh Eppendorf and stored in −80°C until processing for GC-MS analysis. For GC-MS analysis, 10 µl of 0.05 mM norvaline was added to 100 µl of extracted metabolites and dried under $N_2$ gas-flow at 37°C using an evaporator. Amino and organic acids were derivatized via methoximation and silylation. In brief, metabolites were resuspended in 25 µl of methoxylamine hydrochloride (2% [w/v] in pyridine) and incubated at 40°C for 90 min on a heating block. After brief centrifugation, 35 µl of MTBSTFA + 1% TBDMS was added and the samples were incubated at 60°C for 30 min. GC-MS analysis was performed on an Agilent 7890B GC system equipped with a HP-5MS capillary column connected to an Agilent 5977A Mass Spectrometer 50. Mass isotopomer distributions were obtained by integration of ion chromatograms 51 and corrected for natural isotope abundances Relative levels of metabolites or isotopomers were calculated by normalizing to norvaline. MPE of isotopes, an index of isotopic enrichment of metabolites, was calculated as the percent of all atoms within the metabolite pool that are labeled according to the established formula (*Wallace et al., 2018*).

## RNA preparation and quantitative RT-PCR
Total RNA was extracted from tissue or cells using RNeasy mini-kit (Qiagen), and cDNA was synthesized using iScript cDNA Synthesis kit (BioRad) according to the manufactural protocols. qRT-PCR was performed using an ABI ViiA7 PCR cycler. The primer sequences are listed in *Supplementary file 1*. The gene expression levels were normalized to internal control *36B4* levels (*Yoneshiro et al., 2019*).

## Immunoblotting
Protein lysates from isolated BAT were extracted using Qiagen TissueLyzer LT and RIPA lysis and extraction buffer (Thermo Fisher) and cOmplete protease inhibitors (Roche). Tissue lysates were applied to immunoblot analysis using the UCP1 antibody (Amcam, ab10983, 1:2000). β-Actin (Sigma, A3854, 1:40,000) was used as a loading control for each sample.

## Statistical analyses
All data were expressed as the means with SEMs and analyzed by using statistical software (SPSS 25.0; IBM Japan, Tokyo, Japan). Comparisons between the two groups were analyzed using the paired *t*-test or the Student's *t*-test, as appropriate. Analysis of variance (ANOVA) followed by Tukey's test was used for multiple-group comparisons. Two-way repeated-measures ANOVA was used for the comparisons of repeated measurements. Two-tailed p value $\leq$ 0.05 was considered as statistically significant.

## Acknowledgements
We thank Olga Byakina and Yasuo Oguri for their technical support. This work was supported by the NIH (DK097441, DK126160, DK125281, and DK127575 to SK, 5K08HL135275 to RWM, and F32HL137398 to SBC), an American Diabetes Association Pathways to Stop Diabetes Initiator Award (# 1-16-INI-17) to PJW, the Edward Mallinckrodt, Jr. Foundation to SK, MEXT KAKENHI (15H05932, 15K21744, and 20H03418 to KN and 19K06954 to NK), AMED (JP21gm5010002s0305) to KN, and JST Moonshot R&D (JPMJMS2023) to KN. TY was supported by the JSPS Fellowships and the Uehara Memorial Foundation, and JMW was supported by Cardiovascular-Metabolic Fellowship (#1-21-CMF-005).

## Additional information

### Funding

| Funder | Grant reference number | Author |
|---|---|---|
| National Institute of Diabetes and Digestive and Kidney Dis- | DK097441 | Shingo Kajimura |

| eases | | |
| --- | --- | --- |
| National Institute of Diabetes and Digestive and Kidney Diseases | DK126160 | Shingo Kajimura |
| National Institute of Diabetes and Digestive and Kidney Diseases | DK125281 | Shingo Kajimura |
| National Institute of Diabetes and Digestive and Kidney Diseases | DK127575 | Shingo Kajimura |
| Edward Mallinckrodt, Jr. Foundation | | Shingo Kajimura |
| National Heart and Lung Institute | 5K08HL13527 | Robert W McGarrah |
| National Heart and Lung Institute | F32HL137398 | Scott B Crown |
| American Diabetes Association | 1-16-INI-17 | Phillip J White |
| Japan Society for the Promotion of Science | 19K06954 | Kazuhiro Nakamura |
| Japan Society for the Promotion of Science | 15H05932 | Naoya Kataoka |
| Japan Society for the Promotion of Science | 15K21744 | Naoya Kataoka |
| Japan Society for the Promotion of Science | 20H03418 | Naoya Kataoka |
| AMED | JP21gm5010002s0305 | Kazuhiro Nakamura |
| AMED | JPMJMS2023 | Kazuhiro Nakamura |

The funders had no role in study design, data collection and interpretation, or the decision to submit the work for publication.

### Author contributions
Takeshi Yoneshiro, Data curation, Formal analysis, Validation, Investigation, Visualization, Methodology, Writing - original draft; Naoya Kataoka, Resources, Data curation, Validation, Investigation, Methodology; Jacquelyn M Walejko, Data curation, Formal analysis, Investigation; Kenji Ikeda, Resources, Investigation; Zachary Brown, Momoko Yoneshiro, Resources, Validation; Scott B Crown, Data curation, Methodology; Tsuyoshi Osawa, Juro Sakai, Resources; Robert W McGarrah, Phillip J White, Data curation, Formal analysis, Supervision; Kazuhiro Nakamura, Data curation, Formal analysis, Supervision, Validation, Investigation; Shingo Kajimura, Conceptualization, Formal analysis, Supervision, Funding acquisition, Writing - original draft, Project administration

### Author ORCIDs
Takeshi Yoneshiro ⓘ https://orcid.org/0000-0001-5298-5831
Shingo Kajimura ⓘ https://orcid.org/0000-0003-0672-5910

### Ethics
Animal experimentation: All the animal experiments were performed following the guidelines by the UCSF Institutional Animal Care and Use Committee or by the Nagoya University Animal Experiment Committee. The protocols were approved by the committees by the Committee on the Ethics of Animal Experiments of UCSF (AN165833) and Nagoya University. All surgery was performed under anesthesia, and every effort was made to minimize suffering.

### Decision letter and Author response
Decision letter https://doi.org/10.7554/eLife.66865.sa1
Author response https://doi.org/10.7554/eLife.66865.sa2

# Additional files

## Supplementary files
- Supplementary file 1. Primer sequences.
- Transparent reporting form

## Data availability

All data generated or analyzed during this study are included in the manuscript as source data files.

The following previously published datasets were used:

| Author(s) | Year | Dataset title | Dataset URL | Database and Identifier |
|---|---|---|---|---|
| Christian M, Rosell M, Frontini A, Cinti S, Kaforou M, Montana G | 2014 | Expression data from exposure of BAT and WAT at 6 and 28 degrees C | https://www.ncbi.nlm.nih.gov/geo/query/acc.cgi?acc=GSE51080 | NCBI Gene Expression Omnibus, GSE51080 |
| Tajima K, Ikeda K, Chang HY, Chang CH, Yoneshiro T, Oguri Y, Jun H, Wu J, Ishihama Y, Kajimura S | 2019 | RNA-seq of age-associated transcriptome changes in brown adipose tissue | https://www.ebi.ac.uk/arrayexpress/experiments/E-MTAB-7445/ | ArrayExpress, E-MTAB-7445 |
| Wu C, Orozco C, Boyer J, Leglise M, Goodale J, Batalov S, Hodge CL, Haase J, Janes J, Huss JW, Su AI | 2009 | BioGPS: an extensible and customizable portal for querying and organizing gene annotation resources | http://biogps.org/#goto=welcome | BioGPS, GeneAtlas MOE430, gcrma |

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
