## [Decision Letter]

**Acceptance summary:**

Febrile responses require thermogenesis by the brown adipose tissue, but the fuels used to generate heat in this context are incompletely understood. Here the authors identify transport of branched chain amino acids into the mitochondria as an essential component of febrile responses to pyrogens and physiological stress in rodents. Genetic deletion of the mitochondrial branched chain amino acid transporter SLC25A44 from brown adipose tissue blunts these responses and alters nutrient utilization in brown adipocytes.

**Decision letter after peer review:**

Thank you for submitting your article "Metabolic flexibility via mitochondrial BCAA carrier SLC25A44 is required for optimal fever" for consideration by *eLife*. Your article has been reviewed by 3 peer reviewers, including Ralph J DeBerardinis as the Reviewing Editor and Reviewer #1, and the evaluation has been overseen by David James as the Senior Editor.

Essential revisions:

1. Where is the level of BCAA regulation after PGE_2_ or CL treatment? Does this occur via enhanced uptake across the plasma membrane or enhanced uptake into the mitochondria? Does BCAA uptake in isolated mitochondria increase after treatment with PGE_2_ or β-3-AR agonists?

2. In interpreting the data on the PGE_2_-mediated thermogenic response, it would be helpful to know whether UCP1 protein expression is increased in the brown adipose tissue.

3. Regarding the data in Figure 4F and 5A-D, the authors write "However, norepinephrine-induced thermogenesis was significantly enhanced when Leu was supplemented even in the presence of high glucose at 20 mM (Figure 5C). Importantly, this effect of Leu was not seen in the absence of MBC (Figure 5D)." This is confusing: In Figure 5C and D, Val is shown. Can the authors directly compare the O2 consumption response of MBC-KO adipocytes to the three amino acids individually – valine, leucine and isoleucine (with and without high glucose) – administered during NE activation?

4. The relevance of the enhanced glycolysis observed in Figure 5E is unclear. Converting glucose to lactate should not support thermogenesis if UCP1 and therefore oxidative mitochondrial metabolism is required. Does NE stimulation induce glucose oxidation in addition to increased lactate formation under these conditions, and is this effect further enhanced in MBC-deficient BAT?

*Reviewer #1:*

This manuscript describes the role of BCAA metabolism in PGE_2_-mediated fever. The authors find that BAT-specific knockout of the mitochondrial BCAA carrier SLC25A44 (MBC) results in blunted thermogenesis following intracerebroventricular PGE_2_ administration. The febrile state is associated with increased expression BCAA-related genes including MBC in BAT and marked depletion of BCAAs from the circulation. BAT-specific knockout of BCKDHA, which catalyzes the rate-limiting step of mitochondrial BCAA oxidation, also blunts PGE_2_-mediated fever. Isolated BAT tissue from mice exposed to PGE_2_ or psychological stress reveals enhanced BCAA oxidation, and providing excess BCAAs in these culture-based experiments allows BAT to sustain respiration stimulated by norepinephrine. Metabolomics and stable isotope tracing studies indicate that MBC knockout results in suppressed pools of TCA cycle intermediates and some nonessential amino acids. The authors conclude that MBC-mediated BCAA oxidation is required to fuel ongoing thermogenesis during febrile responses. Overall, the experiments are performed to a high standard, make use of genetic mouse models, and support a clear narrative of the role of MBC in thermogenesis stimulated by central PGE_2_ administration.

1. As the authors point out in the Discussion, the metabolic effects of BCAA degradation can include anaplerosis and/or the production of acetyl-CoA. This makes the respiration experiments in Figure 5 difficult to interpret, because Leu (used in Figure 5A,B) generates acetyl-CoA, whereas Val (Figure 5C,D) is anaplerotic. The effects on respiration may therefore differ. Can the authors harmonize these experiments by using the same amino acid for all conditions?

2. The relevance of the enhanced glycolysis observed in Figure 5E is unclear. Converting glucose to lactate should not support thermogenesis if UCP1 and therefore oxidative mitochondrial metabolism is required. Does NE stimulation induce glucose oxidation in addition to increased lactate formation under these conditions, and is this effect further enhanced in MBC-deficient BAT?

3. In Figure 6H, labeling in TCA cycle intermediates expressed as MPE% indicates extremely low labeling (under 1%). I may be misinterpreting the data, but the authors should explain how such a small contribution to the TCA cycle can explain a relatively large increase in respiration. Was the precursor pool (leucine, in this case) only partially labeled?

*Reviewer #2:*

This work provides a new understanding of how PGE_2_, a known pyrogen, impacts branched chain amino acid metabolism. There is little known about the metabolic adaptations that organisms undertake when a fever response is activated by PGE_2_. The authors used sophisticated in vivo approaches to administer PGE_2_ through ICV administration while simultaneously measuring body temperature. PGE_2_ treatment led to depletion of serum BCAA levels, while influx of BCAAs into BAT increased. Mitochondrial BCAA uptake via MBC is required for optimal thermogenesis induced by PGE_2_. The authors show that activated brown fat increases BCAA catabolism and provides carbons that provide substrate for the TCA cycle. The use of steady state and isotopic tracer analysis provides strong evidence for BCAA metabolism in fever-induced thermogenesis, which at the moment is poorly understood.

This is a well written manuscript that addresses the impact of PGE_2_ on BCAA metabolism. The authors provide a thorough analysis of PGE_2_ action in vivo. There are a few suggestions to help round out the paper and provide some clarity.

1. Does BCAA uptake in isolated mitochondria increase with PGE_2_ or CL treatment?

2. Where is the regulation of BCAA uptake? Is it by increasing uptake across plasma membrane or uptake in mitochondria?

3. Although loss of MBC decreases uptake of BCAA, there still remains substantial amount of Valine oxidation in knockout mice. Would that suggest that there is another mitochondrial BCAA transporter?

4. Does PGE_2_ increase UCP1 protein expression? A lot of the analysis includes changes in gene expression. However, protein expression has not been addressed.

*Reviewer #3:*

The authors set out to study the role of branched chain amiono acids (BCAA) into brown adipose tissue (BAT) during fever. BAT is a special type of fat that is important for non-shivering thermogenesis as a defence against a cold environment. Using BAT-specific knockout of BCAA and sophisticated models of fever, they show that the BCAA transporter MBC is important for the febrile response.

Yoneshiro et al. report that mitochondrial BCAA oxidation in brown adipose tissue (BAT) is significantly enhanced during fever. Genetic deletion of MBC in a BAT-specific manner reduces BAT function following intracerebroventricular PGE_2_ administration. The BCAA-induced increase in O2 consumption is blunted in MBC-null adipocytes.

This is a very interesting study and I have only one (rather minor) point that should be addressed:

Effect of BCAA on adipocytes: Figure 4f and 5a-d: The authors write "However, norepinephrine-induced thermogenesis was significantly enhanced when Leu was supplemented even in the presence of high glucose at 20 mM (Figure 5C). Importantly, this effect of Leu was not seen in the absence of MBC (Figure 5D)." This is confusing: In Figure 5C and D Val is shown. It would be interesting to directly compare the response (O2 consumption) of MBC-KO adipocytes to the three amino acids valine, leucine and isoleucine (with and without high glucose) administered during NE activation.

---

## [Author Response]

Reviewer #1:[…] 1. As the authors point out in the Discussion, the metabolic effects of BCAA degradation can include anaplerosis and/or the production of acetyl-CoA. This makes the respiration experiments in Figure 5 difficult to interpret, because Leu (used in Figure 5A,B) generates acetyl-CoA, whereas Val (Figure 5C,D) is anaplerotic. The effects on respiration may therefore differ. Can the authors harmonize these experiments by using the same amino acid for all conditions?

We appreciate the reviewer’s comment. We agree that this is an important point. To address the comment, we tested if supplementation of three BCAAs (Valine, Leucine, and Isoleucine) stimulates respiration in brown adipocytes. We found that all of them significantly augmented norepinephrine (NE)-stimulated respiration in an MBC-dependent fashion (see **New Figure 5C**).

The data suggest that the role of BCAA in thermogenic fat is likely more than merely a carbon source into the TCA cycle. This includes mitochondrial nitrogen supply, such as the synthesis of mitochondrial amino acids (Glu, Ala, Asp), given the fact that all the three BCAAs are nitrogen donors through the common deamination step by BCAT_2_. We discussed this point on Page 11 and 16.

2. The relevance of the enhanced glycolysis observed in Figure 5E is unclear. Converting glucose to lactate should not support thermogenesis if UCP1 and therefore oxidative mitochondrial metabolism is required. Does NE stimulation induce glucose oxidation in addition to increased lactate formation under these conditions, and is this effect further enhanced in MBC-deficient BAT?

We apologize for the confusion. It is well known that β-adrenergic receptor (β-AR) stimulation through norepinephrine (NE) induces both glucose oxidation and lactate production in brown adipocytes (Olsen et al., 2014, Ikeda et al., 2017). We measured lactate production as a readout to assess the fuel choice between glucose vs. BCAA in brown adipocytes following NE stimuli.

The data of higher lactate production in MBC KO cells relative to control cells (**Figure 5F**) suggest that MBC-deficient cells are limited to utilize glucose for NE-induced thermogenesis even in the presence of Leu. We discussed this point in our revised manuscript (Page 11).

3. In Figure 6H, labeling in TCA cycle intermediates expressed as MPE% indicates extremely low labeling (under 1%). I may be misinterpreting the data, but the authors should explain how such a small contribution to the TCA cycle can explain a relatively large increase in respiration. Was the precursor pool (leucine, in this case) only partially labeled?

Thank you for raising this issue. This is due to the fact that the present study employed a shorter labeling time (30 min) relative to our previous study that achieved ~20% MPE of TCA intermediates (Yoneshiro et al., 2019). In the present study, we aimed to determine how MBC loss affects the downstream pathway of Leu catabolism following NE stimulation. To this end, the cells were incubated with [^13^C]-Leu-containing medium for 30 min, and subsequently washed off (nonlabeled medium) and stimulated with NE for 1 hour (see **Figure 6**－**figure supplement 1A**). In contrast, our previous work (Yoneshiro et al., 2019) used a protocol in which cells were labeled for 1 hour during NE treatment. We discussed this point in our revised manuscript (Page 13).

Reviewer #2:[…] This is a well written manuscript that addresses the impact of PGE_2_ on BCAA metabolism. The authors provide a thorough analysis of PGE_2_ action in vivo. There are a few suggestions to help round out the paper and provide some clarity.1. Does BCAA uptake in isolated mitochondria increase with PGE_2_ or CL treatment?

We thank the reviewer for the constructive comments. Both CL treatment and PGE_2_ increased BCAA oxidation in the BAT (**Figures 1F, 2B).** Consistent with the data, we demonstrated that norepinephrine (NE) treatment increased the BCAA flux in the mitochondria (**Figure 5A, 5C**, and also in Yoneshiro et al., 2019). Hence, imported BCAAs to the isolated mitochondria would be rapidly oxidized (*i.e.,* the mitochondrial BCAA content would be reduced) when stimulated by PGE_2_ or CL.

In fact, we found that isolated mitochondria from NE-stimulated brown adipocytes have lower [^14^C]-labeled Leu contents relative to non-stimulated brown adipocytes, due to high BCAA oxidation rate/flux (see Author response image 1). We would need to use a non-metabolized BCAA tracer to measure the accumulated Val contents in the mitochondria.

**Author response image 1. respfig1:** Effects of norepinephrine (NE) on mitochondrial BCAA content in brown adipocytes. Differentiated brown adipocytes were treated with NE for 20 min and harvested to isolate mitochondria of brown adipocytes with shRNA knockdown of MBC (KD) or control cells. Subsequently, isolated mitochondria were incubated with [U-14C] Val at 37°C for 1 h. After washing with PBS 3 times, [U-14C] Val content in mitochondria was quantified by using a scintillation counter. n = 3/group. ANOVA with Tukey’s HSD test.

2. Where is the regulation of BCAA uptake? Is it by increasing uptake across plasma membrane or uptake in mitochondria?

This is an important point. Our recent study found that MBC overexpression increased the expression of SLC7A5 (LAT1), the plasma membrane transporter for neutral amino acids, including BCAAs (Walejko et al., 2021). The data suggest that mitochondrial BCAA uptake via MBC is linked to the plasma membrane uptake, although the underlying mechanism remains unknown.

We are also aware that BCAA oxidation is rapidly induced by norepinephrine or CL stimuli within a few minutes (see **Figure 5A**). It is possible that post-translational modifications, such as phosphorylation of MBC, may control MBC activity in a signal-dependent fashion. Thus, our future studies will investigate the signal-dependent post-translational modification of MBC.

We discussed this in our revised manuscript (Page 16).

3. Although loss of MBC decreases uptake of BCAA, there still remains substantial amount of Valine oxidation in knockout mice. Would that suggest that there is another mitochondrial BCAA transporter?

This is certainly possible. However, it is important to note that the BAT-specific KO mice lack MBC in brown adipocytes but not in other UCP1-negative cell types, such as endothelial cells, within the BAT. Thus, it is possible that Valine oxidation in KO mice may arise from UCP1negative cells in the BAT. We discussed this point on Page 8.

Another possibility is the contribution of keto-acids, *e.g.,* α-ketoisovalerate (KIV). We previously showed that KIV uptake is not affected by MBC deletion (Yoneshiro et al., 2019). Although the expression of BCAT1 (the cytosolic form of BCAT) is low in BAT, oxidation of labeled KIV (deaminated from labeled Valine via BCAT1) could contribute to Valine oxidation.

4. Does PGE_2_ increase UCP1 protein expression? A lot of the analysis includes changes in gene expression. However, protein expression has not been addressed.

This is a valid point. Our new data showed that UCP1 protein expression was unchanged after 2 hours of an ICV administration of PGE_2_ (**New Figure 1D).**

Reviewer #3:[…] This is a very interesting study and I have only one (rather minor) point that should be addressed:Effect of BCAA on adipocytes: Figure 4f and 5a-d: The authors write "However, norepinephrine-induced thermogenesis was significantly enhanced when Leu was supplemented even in the presence of high glucose at 20 mM (Figure 5C). Importantly, this effect of Leu was not seen in the absence of MBC (Figure 5D)." This is confusing: In Figure 5C and D Val is shown. It would be interesting to directly compare the response (O2 consumption) of MBC-KO adipocytes to the three amino acids valine, leucine and isoleucine (with and without high glucose) administered during NE activation.

We appreciate the reviewer’s comment. We agree that this is an important point. To address the comment, we tested if supplementation of three BCAAs (Valine, Leucine, and Isoleucine) stimulates respiration in brown adipocytes. We found that all of them significantly augmented norepinephrine (NE)-stimulated respiration in an MBC-dependent manner (**New Figure 5C**).

The data suggest that the role of BCAA in thermogenic fat is more than merely a carbon source of fuel into the TCA cycle. This includes mitochondrial nitrogen supply, including the synthesis of mitochondrial amino acids (Glu, Ala, Asp), given the fact that all the three BCAAs are major nitrogen donors through the common deamination step by BCAT2. We discussed this point on Page 11 and 16.

**References:**

Ikeda, K., Kang, Q., Yoneshiro, T., Camporez, J. P., Maki, H., Homma, M., Shinoda, K., Chen, Y., Lu, X., Maretich, P., Tajima, K., Ajuwon, K. M., Soga, T. and Kajimura, S. 2017. UCP1-independent signaling involving SERCA2b-mediated calcium cycling regulates beige fat thermogenesis and systemic glucose homeostasis. *Nat Med,* 23, 1454-1465.

Olsen, J. M., Sato, M., Dallner, O. S., Sandstrom, A. L., Pisani, D. F., Chambard, J. C., Amri, E. Z., Hutchinson, D. S. and Bengtsson, T. 2014. Glucose uptake in brown fat cells is dependent on mTOR complex 2-promoted GLUT1 translocation. J Cell Biol, 207, 365-74.

Pedersen, H. K., Gudmundsdottir, V., Nielsen, H. B., Hyotylainen, T., Nielsen, T., Jensen, B. A., Forslund, K., Hildebrand, F., Prifti, E., Falony, G., Le Chatelier, E., Levenez, F., Dore, J., Mattila, I., Plichta, D. R., Poho, P., Hellgren, L. I., Arumugam, M., Sunagawa, S., Vieira-Silva, S., Jorgensen, T., Holm, J. B., Trost, K., Meta, H. I. T. C., Kristiansen, K., Brix, S., Raes, J., Wang, J., Hansen, T., Bork, P., Brunak, S., Oresic, M., Ehrlich, S. D. and Pedersen, O. 2016. Human gut microbes impact host serum metabolome and insulin sensitivity. *Nature,* 535, 376-81.

Walejko, J. M., Christopher, B. A., Crown, S. B., Zhang, G. F., Pickar-Oliver, A., Yoneshiro, T., Foster, M. W., Page, S., Van Vliet, S., Ilkayeva, O., Muehlbauer, M. J., Carson, M. W., Brozinick, J. T., Hammond, C. D., Gimeno, R.

E., Moseley, M. A., Kajimura, S., Gersbach, C. A., Newgard, C. B., White, P. J. and Mcgarrah, R. W. 2021. Branched-chain alpha-ketoacids are preferentially reaminated and activate protein synthesis in the heart. *Nat Commun,* 12, 1680.

Wolfe, R. R. 2017. Branched-chain amino acids and muscle protein synthesis in humans: myth or reality? *J Int Soc Sports Nutr,* 14, 30.

Yoneshiro, T., Wang, Q., Tajima, K., Matsushita, M., Maki, H., Igarashi, K., Dai, Z., White, P. J., Mcgarrah, R. W., Ilkayeva, O. R., Deleye, Y., Oguri, Y., Kuroda,

M., Ikeda, K., Li, H., Ueno, A., Ohishi, M., Ishikawa, T., Kim, K., Chen, Y., Sponton, C. H., Pradhan, R. N., Majd, H., Greiner, V. J., Yoneshiro, M., Brown, Z., Chondronikola, M., Takahashi, H., Goto, T., Kawada, T., Sidossis, L., Szoka, F. C., Mcmanus, M. T., Saito, M., Soga, T. and Kajimura, S. 2019. BCAA catabolism in brown fat controls energy homeostasis through SLC25A44. *Nature,* 572, 614-619.